# The Feasibility of Using Pulsed-Vacuum in Stimulating Calcium-Alginate Hydrogel Balls

**DOI:** 10.3390/foods10071521

**Published:** 2021-07-01

**Authors:** Janjira Jinnoros, Bhundit Innawong, Patchimaporn Udomkun, Pramuk Parakulsuksatid, Juan L. Silva

**Affiliations:** 1Department of Food Technology, Faculty of Engineering and Industrial Technology, Silpakorn University, Nakhon Pathom 73000, Thailand; b_innawong@yahoo.com; 2Silpakorn University Food Innovation Hub (SUFIH), Pathum Thani 12120, Thailand; 3International Institute of Tropical Agriculture (IITA), Bujumbura P.O. Box 1893, Burundi; p.udomkun@cgiar.org; 4Department of Biotechnology, Faculty of Agro-Industry, Kasetsart University, Bangkok 10900, Thailand; fagipmp@ku.ac.th; 5Department of Food Science, Nutrition and Health Promotion, Mississippi State University, Starkville, MS 39762, USA; jsilva@foodscience.msstate.edu

**Keywords:** calcium diffusion, depressurization, external gelation, gel structure, gelation behavior, hydrodynamic mechanisms, vacuum pulse

## Abstract

The effect of the pulsed-vacuum stimulation (PVS) on the external gelation process of calcium-alginate (Ca-Alg) hydrogel balls was studied. The process was conducted at four different working pressures (8, 35, 61, and 101 kPa) for three pulsed-vacuum cycles (one cycle consisted of three repetitions of 10 min of depressurization and 10 min of vacuum liberation). The diffusion coefficients (D) of calcium cations (Ca^2+^) gradually reduced over time and were significantly pronounced (*p* < 0.05) at the first three hours of the external gelation process. The rate of weight reduction (WR) and rate of volume shrinkage (S_v_) varied directly according to the D value of Ca^2+^. A significant linear relationship between WR and S_v_ was observed for all working pressures (*R^2^* > 0.91). An application of a pulsed vacuum at 8 kPa led to the highest weight reduction and shrinkage of Ca-Alg hydrogel samples compared to other working pressures, while 61 kPa seemed to be the best condition. Although all textural characteristics (hardness, breaking deformation, Young’s modulus, and rupture strength) did not directly variate by the level of working pressures, they were likely correlated with the levels of WR and S_v_. Scanning electron micrographs (SEM) supported that the working pressure affected the characteristics of Ca-Alg hydrogel structure. Samples stimulated at a working pressure of 8 kPa showed higher deformation with heterogenous structure, large cavities, and looser layer when compared with those at 61 kPa. These results indicate the PVS is a promising technology that can be effectively applied in the external gelation process of Ca-Alg gel.

## 1. Introduction

Alginates or alginic acid, are unbranched polysaccharides isolated from marine brown algae such as Laminaria sp., Durvillaea sp., and Sargassum sp. [1] or bacterial biofilms [2]. Alginate polymer is composed of (1–4)-β-D-mannuronic acid (M) and α-L-guluronic acid (G) sugar monomers, which can be separated into three groups: M-blocks, G-blocks, and a mixture of M and G residues [1,3]. The composition of M and G in alginate depends on the source of algae and bacteria [2]. Alginate has a carboxyl group with a high ion-exchange property. Thus, alginate can form a metal salt with an alkali metal ion (monovalent ions) such as Na^+^ and K^+^ and multivalent ions such as Ca^2+^ [4]. Due to its capability of forming gel networks in the presence of cations, particularly Ca^2+^, alginates have been applied in various areas such as food additives in the food industry and encapsulation agents in biotechnology and pharmacology [5,6,7,8].

In general, Ca^2+^ selectively binds to G-block of alginates and then forms a crosslinked gel network [9]. When Ca^2+^ binds four G units, a hexagonal lattice, which is known as an “egg-box” gel structure, is formed [10]. By this, alginate polymers with a high number of G-blocks can form more ionic crosslinks and more inflexible gels [11]. However, Lozano-Vazquez et al. [12] indicated that the addition of calcium in the system plays an important role in inducing intermolecular ionic bonds between polymeric chains of alginate for the formations of gels. Presently, Ca^2+^-crosslinked alginate gels (calcium alginate; Ca-Alg) with growth factors, cells, and/or cytokines encapsulations have been used in vivo for a vast number of applications such as in diabetes treatment [13], bone regeneration [14], and tissue repair [15]. Moreover, a crosslink of alginate-based hydrogels is also used as an effective tool for biomedical sensing [16] and controlling the drug delivery process [17].

Ca-Alg gel can be produced by external and internal gelation methods. In the external gelation process, the Ca-alginate gel particles can be easily produced by extruding the alginate solution into a calcium salt solution, leading to instantaneous gelation of the alginate [18]. In this method, Ca^2+^ diffuses from the solution to the interior of the alginate droplets, forming a Ca-Alg gel matrix from the outside to the center [19]. In the internal gelation method, an insoluble calcium salt (i.e., CaCO_3_ or CaSO_4_) is added to the alginate solution and the mixture is extruded into oil [20]. Subsequently, acids (i.e., acetic acid or glucono-delta-lactone/GDL) are used to release Ca^2+^ from the insoluble salt for crosslinking with the alginate. Although some studies have reported that the Ca-Alg gel produced by the internal gelation method was more homogeneous [20,21], the internal gelated matrices were less dense with larger pore sizes, resulting in lower encapsulation efficiency and faster release rate [22]. Chan et al. [18] investigated the effect of internal and external gelation methods on the properties of alginate film. The result showed that external crosslinking produced films with a thinner and smoother surface, greater matrix strength, stiffness, and permeability than internally crosslinked films. The external gelation method is the preferred method for producing crosslinked alginate for coating and encapsulation purposes. The properties of alginate gel depend on both intrinsic factors (e.g., the molecular weight, Ca-binding blocks distribution, or the degree of methoxylation) and extrinsic factors (e.g., polyuronates concentration, pH, temperature, or ion strength) [10]. The diffusion rate of cations during the external gelation process has also been reported as a limiting factor in alginate hydrogel production [23].

Indeed, few studies have been performed to improve the diffusion rate of cations in the production of alginate hydrogel droplets and microparticles. For example, Bajpai et al. [24] prepared Ca(II)+Ba(II) ions crosslinked alginate hydrogels by a novel diffusion through the dialysis tube (DTDT) technique, while Simonescu et al. [25] prepared Ca-Alg beads for use in removing lead from natural and wastewaters by ion exchange using the ultrasound technique. The results showed that the beads prepared using ultrasound showed a greater ion exchange capability. Vacuum infiltration is a technique frequently used in plant biochemistry research. It is performed by dipping a tissue into a solution that is subsequently brought to a vacuum condition [26]. The vacuum induces the expulsion of the naturally occurring interstitial air between the cells in the plant matrices. Then, the solution in which plant tissue is immersed could penetrate along with the solutes in the tissue structure without damaging cells [27]. In food science and technology, this technique is well known as vacuum impregnation or vacuum infusion. Vacuum impregnation has been used to add biologically active substances into food to improve the nutritional value, shelf-life, and texture of agricultural products [28,29,30,31,32]. Servillo et al. [26] employed this technique to allow calcium ions to penetrate more efficiently in the tissue structure of diced tomato with a consequent increase in product firmness. Although the application of a vacuum pulse technique showed high potential in increasing the ion incorporation in food products, no studies regarding the effect of pulsed-vacuum stimulation (PVS) through the external gelation process on the properties of Ca-Alg hydrogel were published. Therefore, the objective of this study was to investigate the feasibility of using PVS at different depressurization levels on the diffusion coefficients of calcium ion, textural and microstructural characteristics of Ca-Alg hydrogel balls. The findings in the current work provide a better understanding of the effect of PVS on the quality attributes of hydrogel ball and provide a new approach for the development of novel functional restructured products.

## 2. Materials and Methods

### 2.1. Chemicals

Sodium alginate was purchased from Topflight Inter Foods Co., Ltd. (Bangkok, Thailand). Dihydrate calcium chloride (CaCl_2_, >95% purity) was provided by Thermo Fisher Scientific Australia Pty Ltd. (Victoria, Australia). Others were of analytical grade and were used without further purification.

### 2.2. Formation of Alginate Balls

In this study, a solution containing 1% (*w*/*w*) sodium alginate was prepared by mixing alginate with deionized water under intense stirring until complete dissolution. Before hydrogel preparation, the alginate solution was left to stand at room temperature for 24 h to remove air bubbles. After that, 15 g alginate solution was weighed and then formed a spherical shape using a plastic mold. The alginate balls of 3 × 3 cm^2^ were stored in the freezer at −18 °C for 12 h prior to forming Ca-Alg hydrogels.

### 2.3. Preparation of Calcium-Alginate Hydrogel Balls

Before applying a PVS process, the externally crosslinked Ca-Alg balls were prepared by soaking alginate balls in 3% *(w/w)* CaCl_2_ solution for 15 min at room temperature (15 balls per 2 L CaCl_2_) to form a thin gel layer. This process was required as the direct application of the PVS process could damage the Ca-Alg hydrogel balls.

### 2.4. Pulsed-Vacuum Stimulation Process

A jacketed stainless-steel vacuum chamber (Owner Foods Machinery Co., Ltd., Bangkok, Thailand) was used in preparing Ca-Alg hydrogel. The chamber with a capacity of 8 L (diameter 19.8 cm and height 28 cm) was connected to a vacuum pump. In this study, three different depressurization levels (8, 35, and 61 kPa) of the PVS process were applied to Ca-Alg hydrogel balls for 3 cycles (180 min). One cycle of PVS consisted three repetitions of depressurization for 10 min and liberation for 10 min (60 min per cycle). The control sample was carried out at 101 kPa without subjecting it to the PVS process. Ca-Alg hydrogels were rinsed with deionized water before analysis.

### 2.5. Determination of Weight Loss

The weight reduction (WR) of the samples was calculated using Equation (1):(1)$$\mathrm{WR}\left(\%\right)=\frac{\left(W_{0}-W_{t}\right)}{W_{0}}\times 100$$
where W_0_ is the initial weight of the sample (g) and W_t_ is the weight of the hydrogel balls at time t. Six samples were analyzed for each treatment.

### 2.6. Determination of Diffusion Coefficient of Calcium Ion

The method of Chrastil [33] was used to determine the diffusion coefficients (D) of calcium ions with slight modifications. All Ca-Alg ball samples were cut in half, and then the thickness of the gel layer was measured using a caliper. Fifteen samples were analyzed for each treatment. The D value was calculated using Equations (2) and (3) as follows:(2)$$L=L_{\max}{\left[1-\exp\left(-\mathrm{kt}\right)\right]}^{n}$$
where L is the thickness of hydrogel in % at time t, Lmax is the thickness of hydrogel in % at t_→∞_ (the radius of the ball when the gelation is completed), n is the heterogeneous structural resistance constant, and k is the gelation rate constant. The nonlinear regression method was used to compute n and k values.

The diffusion coefficients of the calcium ion were calculated as follows:(3)$$D=\frac{\pi {\left(k\right)}^{2n}{\left(L_{\max}\right)}^{2}}{36}$$

### 2.7. Volume Shrinkage Measurement

Shrinkage in volume (S_v_) was calculated using Equation (4) as follows:(4)$$S_{V}=\frac{V_{0}-V_{t}}{V_{0}}\times 100$$
where V_o_ is the original volume of the hydrogel ball (cm^3^) and V_t_ is the hydrogel volume at time t (cm^3^). Six samples were analyzed for each treatment.

### 2.8. Textural Analysis

A uniaxial compression test at a pretest, test, and post-test speeds of 1, 1, and 10 mm/s, respectively, was conducted to measure the textural characteristic of Ca-Alg hydrogel balls using a texture analyzer (model TA-XT Plus, Stable Micro System Co, Ltd., Surrey, Godalming GU7 1YL, UK). A deformation of 70% was applied. The breaking force or hardness value was calculated from the maximum force (N) required to break the hydrogel. The breaking deformation represented the distance traveled by probe from the hydrogel surface at the point of breakage. The Young’s modulus (E) was computed using the Hertz equation [34] as follows:(5)$$E=\frac{3\left(1-v^{2}\right)F}{4D^{\frac{3}{2}}d^{\frac{1}{2}}}$$
where E is the Young’s modulus, v is Poisson’s ratio of the ball, F is the applied force during the compression (N), D is the deformation (m), and d is the diameter of the hydrogel ball (m). Six samples were tested for each treatment.

Rupture strength was calculated as the penetration force required to break gels [35]. In this study, the rupture strength of Ca-Alg hydrogel balls was also measured using a texture analyzer equipped with a 50 kg load static cell and a flat cylindrical probe (P/2, 2 mm diameter). The pretest, test, and post-test speeds were set at 1, 1, and 10 mm/s, respectively. Six samples were analyzed for each treatment.

### 2.9. Image Acquisition

An image acquisition system consisting of a black wooden chamber (45 × 45 × 45 cm^3^) and a digital camera was assembled. A lightproof chamber was equipped with two parallel fluorescent lamps (TLD90 Deluxe, Natural Daylight, 18W/965, Philips, Holland) with a color temperature of 6500K (D65) and a color rendering index up to 90%. Both lamps were switched on 15 min before image acquisition. A digital camera (Model OMD EM10 Mark II, Olympus, Japan) with a Zuiko 30 mm macro lens was used with a distance between the camera and a sample of 20 cm. The angle between the camera lens and light source axis was set to 45° to capture the diffuse reflection. The camera’s parameters were 1/8 s shutter speed, macrofocusing mode, F3.5 aperture stop, auto ISO sensitivity, and no flash. Images of samples were photographed on a matte black ceramic surface to avoid direct reflection. The angle between the camera lens axis and the sample was 90° to reduce gloss. Six samples were captured for each treatment.

### 2.10. Characterization of Hydrogel Structure

The external and cross-sectional structures of Ca-Alg hydrogel samples were characterized using a scanning electron microscope (model JSM-6610LV, Tokyo, Japan) and energy dispersive X-ray spectrometer (SEM-EDS) (model X-MaxN 50, JEOL Ltd., Tokyo, Japan). The samples were cut to 5 mm^3^ and then dehydrated by immersing in a series of ethanol solutions of increasing concentration until 100% for 30 min each. Subsequently, the samples were immersed in pure ethanol for 60 min before drying in a critical point dryer (model Q150R S, Leica, EM CPD300, Quorum Technology Ltd., East Sussex, England). Samples were then fixed to an aluminum stub using double-sided carbon tape. All samples were then coated with a gold palladium layer under vacuum conditions. The coated samples were photographed using the SEM at an accelerating voltage of 15 kV. Three samples were analyzed for each treatment.

### 2.11. Statistical Analysis

All experiments were replicated three times. The results were expressed as means ± standard deviation. Statistical analysis was performed using SPSS software v. 16 (SPSS Inc., Chicago, IL, USA). Data were analyzed for significant differences by the least significant difference (LSD) test for both experiments. A *p*-value less than 0.05 was considered statistically significant. Linear regression analyses were conducted using OriginPro software v. 9 (OriginLab Corporation, Northampton, MA, USA).

## 3. Results and Discussion

### 3.1. Diffusion Coefficients of Calcium Ion

The results showed that working pressure and depressurization cycle influenced the D value of Ca^2+^ in hydrogel balls (Table 1). At the same working pressure, a significantly higher (*p* < 0.05) D value was recorded at the first three cycles (0–60 min) of the external gelation process with and without PVS application, while the least diffusion of Ca^2+^ was observed in the last cycle (120–180 min). The D value at the first three cycles was about 1.8–11 times higher than the second of the three cycles (60–120 min). This result is similar to the findings of Li et al. [36] and Hajikhani et al. [37]. They stated that initial Ca^2+^ cations, which have a much smaller size than the pore size of alginate gel, could diffuse unhindered into the alginate matrix before solidification.

When considering the first three cycles during the PVS process, the results displayed that the D value at the working pressure of 61 kPa was significantly higher (*p* < 0.05) than 35 and 8 kPa by approximately 1.8 times. Interestingly, the D value at the working pressure of 61 kPa was about 2.2 times lower than 35 kPa and 3.3 times lower than 8 kPa at the second of the three cycles. An increase in the Ca^2+^ cation diffusion at the beginning of the external gelation process with and without PVS application could be attributed to the diffusion mass transfer of solutes in hydrogels [38]. Lee et al. [39] stated that in the external gelation method, the Ca^2+^ cations diffuse inward into the alginate droplet from the gelation bath at a high flow rate due to the high concentration gradient at the interface between the droplet and gelation bath. This mechanism leads to a rapid formation of a layer of a superficial crust at the periphery of the alginate droplet [39,40]. In this study, when more Ca^2+^ cations diffuse into the hydrogel sample, particularly at 61 kPa, more alginate molecules were gelled. The external crust of the hydrogel balls progressively tightened diffusion and reaction between the diffusing CaCl_2_ through alginate solution. Subsequently, the penetration of solutes into the samples is limited, as notified by Lee et al. [39].

When comparing the effect of PVS with the control (101 kPa), a sudden increase in Ca^2+^ diffusion was found, which was significantly pronounced (*p* < 0.05) at the first and second of the three cycles of PVS (Table 1). At the first cycle, the D value of control samples displayed a reduction by 2.6, 1.5, and 1.4% compared to those samples of PVS application at 61, 35, and 8 kPa, respectively. The D value of the control sample continuously decreased at the second cycle compared to pulsed-vacuum stimulated samples which was by approximately 5.4% at 8 kPa, 3.6% at 35 kPa, and 1.6% at 61 kPa. This finding might be ascribed to the action of hydrodynamic mechanisms (HDM) coupled with diffusive phenomena during the pulsed-vacuum process that can promote a higher mass transfer than the atmospheric pressure [29,41,42].

### 3.2. Weight Reduction and Shrinkage

The weight and S_v_ of hydrogel samples decreased by 22.8 and 21.6%, respectively, after soaking alginate balls in CaCl_2_ solution for 15 min. With and without pulsed-vacuum stimulation, a higher rate of WR and S_v_ of Ca-Alg hydrogel samples was detected at the first of the three cycles of the external gelation process (Table 1). The rate of WR and S_v_ continuously decreased until the last cycle, except at the working pressure of 8 kPa, which was dropped at the second of the three cycles before increasing again at the last cycle. This could be explained by the fact that shrinkage increased with the rate of water removal since more stress was induced in the matrix of Ca-Alg hydrogel by contraction. Lee et al. [39] also ascribed the mass loss and the shrinkage of alginate films by gelation time to the crosslink gelation reaction between the excessive Ca^2+^ cations and alginate molecules. During the gelation process, the volume of the hydrogel gradually decreases, whereas water in the hydrogel is squeezed out (syneresis). Consequently, the alginate films shrink further, lose their mass, and become denser [43,44]. The image clearly shows that the crosslink time during the external gelation process with and without PVS led to noticeable changes of volume and the opaque white color of alginate hydrogel (Figure 1).

To compare the effect of working pressures, significantly higher (*p* < 0.05) rates of WR and S_v_ at the first three cycles of the gelation process were found in samples stimulated at a working pressure of 8 kPa, followed by 61, 101, and 35 kPa, respectively (Table 1). Although the highest rate of WR and S_v_ at the last cycle was significantly noticed (*p* < 0.05) at a working pressure of 8 kPa, the rate was nearly zero for samples stimulated at 61 kPa. Specifically, the WR and S_v_ values of hydrogel balls obtained from a working pressure of 61 kPa were about 26.5 and 22.3% lower than samples of 8 kPa (Figure 2 and Figure 3). Due to the pulsed-vacuum process changes that the system undergoes, the HDM, which can occur concomitantly with the deformation-relaxation phenomenon (DRP), highly promotes the physical, mechanical, and microstructural properties of porous solids [45]. Therefore, in the Ca-Alg gel network, the DRP caused by the vacuum pulse might have a greater influence under high vacuum pressure.

Moreover, a significant relationship between WR versus S_v_ of all working pressures expressed a highly linear correlation with *R^2^* > 0.91 (Table 2). The linear behavior can be interpreted such that the structure of Ca-Alg hydrogel was sufficiently elastic to shrink into the space left by the syneresis of water by the crosslink time of all working pressures. Moreover, it could also be observed that the rate of change under atmospheric conditions was about 1.4 times higher than the rate under PVS application. This means a higher volume of shrinkage was found in Ca-Alg hydrogel stimulated with atmospheric pressure when the same level of weight loss was considered.

### 3.3. Textural Characteristics

In this study, the pulsed vacuum resulted in significantly more (*p* < 0.05) hardness, breaking deformation, and Young’s modulus values compared to the atmospheric pressure stimulated sample for three cycles. At the same time, rupture strength was not affected by working pressures (Table 3). However, these textural parameters did not directly vary with the level of working pressures. The greatest changes of hardness, breaking deformation, and Young’s modulus were found in the samples obtained from the working pressure of 8 kPa, followed by 61, 35, and 101 kPa, respectively. Specifically, the levels of hardness, breaking deformation, and Young’s modulus of Ca-Alg hydrogel balls stimulated at 8 kPa were higher by approximately 78.7, 15.1, and 115.6% than the corresponding value of the control sample. This is likely the result of the shrinking and weight reduction effects [46], which take place when PVS is applied.

### 3.4. Microstructural Characteristics

SEM micrographs revealed the networks consisted of polymer strands and voids on external and cross-sectional structures of all Ca-Alg hydrogel balls (Figure 4 and Figure 5). It could be seen that the working pressure operated during the external gelation process led to noticeable changes in the surface and structure characteristics of hydrogel balls. The outer surface of hydrogel balls without the PVS process (control) was smooth and uniform, while the surface of hydrogel balls stimulated by PVS were rough, especially at a working pressure of 8 kPa (Figure 4). Li et al. [36] reported that surface roughness of Ca-Alg films mainly depends on the swelling degree during crosslinking, as the highly swollen films resulted in polymer folding and formed a heterogenous surface. Therefore, it might be hypothesized that the surface heterogeneity of Ca-Alg hydrogel samples could be due to the relatively higher swelling degree during crosslinking by PVS application.

Interestingly, the application of PVS at a depressurization level of 61 kPa could maintain the external structures of a sample when compared with 35 and 8 kPa. The SEM micrographs also showed that the samples obtained from 61 kPa of depressurization level depicted a network of more connected polymer strands with a better homogeneous porous structure and less cell collapse. In comparison, the more collapsed structure with large cavities was found in Ca-Alg hydrogel ball at a working pressure of 8 kPa. A cross-sectional structure was clearly confirmed that the Ca-Alg hydrogel samples obtained from 8 kPa had a looser network structure and larger pore size than those of control, 35, and 61 kPa, respectively (Figure 5). A higher structural change of pulsed-infusion samples than the atmospheric pressure may be related to the different mass transfer parameters and textural changes [47]. When the pressure pulsation is applied, the air initially present in the product expands and partially flows away. Then, after the sudden restoration of the atmospheric pressure, the residual air is compressed again, creating a hydrodynamic drag [26]. More gas expansion and compression cycles could lead to the deformation of structures, shrinkage, and irregular shape [48,49]. Lin et al. [50] found that the microstructural changes in the osmotically dehydrated mango tissues occurred after restoring the atmospheric pressure during the pulsed-vacuum stimulation. They also concluded that the expansion of gases led to an increase in the intercellular space and an alteration of cell membrane permeability. Nonetheless, it was also found that a nonhomogeneous structure and various sizes of cavities were highly observed at the inner structure more than the external surface (Figure 5). Lee et al. [39] and Puguan et al. [51] explained this phenomenon according to the distribution of Ca^2+^ cations and alginate molecules, which was relatively higher at the surface.

## 4. Conclusions

The application of PVS on the external gelation process of Ca-Alg hydrogel balls was investigated. In one cycle of the PVS process, the system’s vacuum condition was interrupted for 10 min, and then the system was maintained at atmospheric pressure for 10 min intermittently until completing 60 min. The Ca-Alg hydrogel balls were continuously stimulated with a pulse vacuum for 3 cycles (180 min). The results showed that the PVS process favored the diffusion coefficients of Ca^2+^, especially when working pressure of 61 kPa was applied for the first of the three cycles. The diffusion coefficients of Ca^2+^ gradually decreased by the pressurization cycle. The lowest WR and shrinkage were observed when the gelation process was accomplished using PVS at 61 kPa. In this condition, the product presented the lowest value of hardness, breaking deformation, Young’s modulus, and rupture strength. When considering the microstructure, large gaps and looser layers inside the hydrogel sample structure were found after PVS at a working pressure of 8 kPa, which were less observed at 61 kPa. This suggests that this product would be quite resistant to deformation if PVS is applied at suitable working pressure. Regarding to this study, a working pressure of 61 kPa under PVS system was the most effective condition for producing Ca-Alg hydrogel ball. Finally, the pulsed-vacuum technology can be considered as a fast process and promising alternatives for the external gelation method of Ca-Alg hydrogel since it enhanced Ca^2+^ diffusion and improved the overall product characteristics compared to atmospheric pressure. Looking to the future opportunities, this study provides a great opportunity for integrating a vacuum pulse in forming fruit alginate interactions. These new product forms help to control the functional properties of the novel, healthy, and value-added restructured fruit products.

## Figures and Tables

**Figure 1 foods-10-01521-f001:**
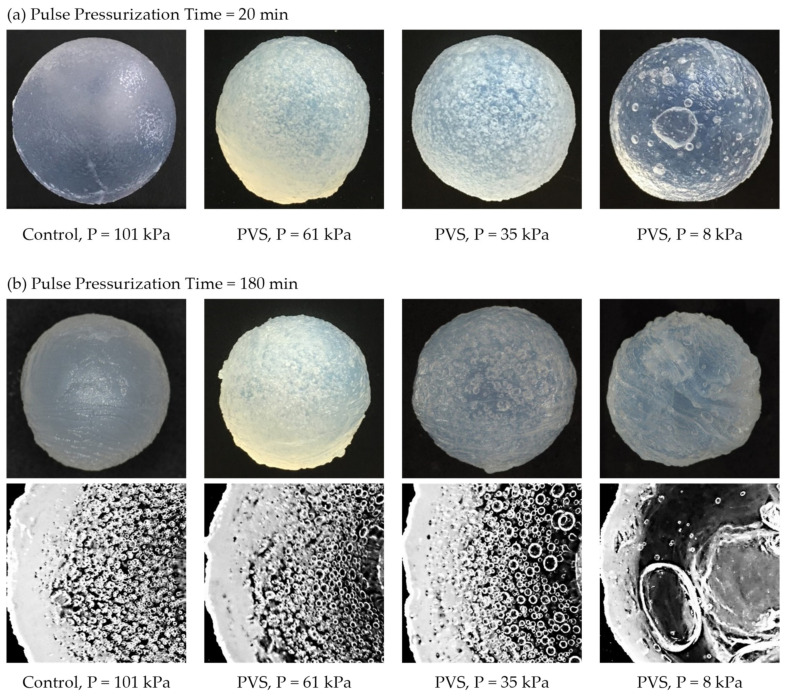
Image of the external surface and cross-sectioned for calcium-alginate hydrogel balls affected by different working pressures and pressurization time: (**a**) Pulse Pressurization Time = 20 min; (**b**) Pulse Pressurization Time = 180 min.

**Figure 2 foods-10-01521-f002:**
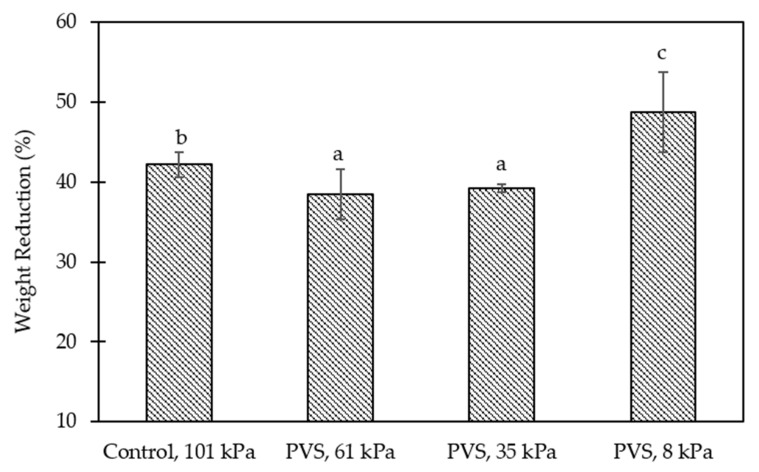
Effect of working pressures on the weight reduction (WR, %) of calcium-alginate hydrogel balls after stimulating for 3 cycles (1 cycle of control = 60 min of soaking during the external gelation process, while 1 cycle of PVS = 10 min of depressurization and 10 min of vacuum liberation for 3 repetitions (60 min)). Means (± standard deviation) within a bar graph followed by different lower-case letters, differ by least significant difference (LSD) test (*p* < 0.05).

**Figure 3 foods-10-01521-f003:**
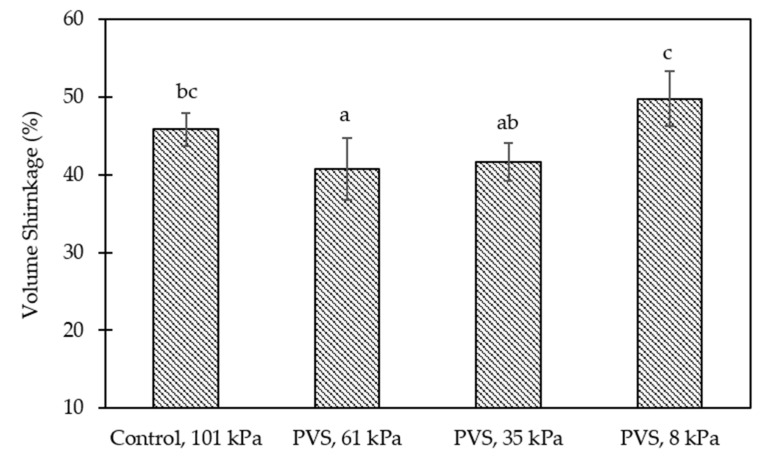
Effect of working pressures on the volume shrinkage (S_v_, %) of calcium-alginate hydrogel balls after stimulating for 3 cycles (1 cycle of control = 60 min of soaking during the external gelation process, while 1 cycle of PVS = 10 min of depressurization and 10 min of vacuum liberation for 3 repetitions (60 min)). Means (± standard deviation) within a bar graph followed by different lower-case letters, differ by least significant difference (LSD) test (*p* < 0.05).

**Figure 4 foods-10-01521-f004:**
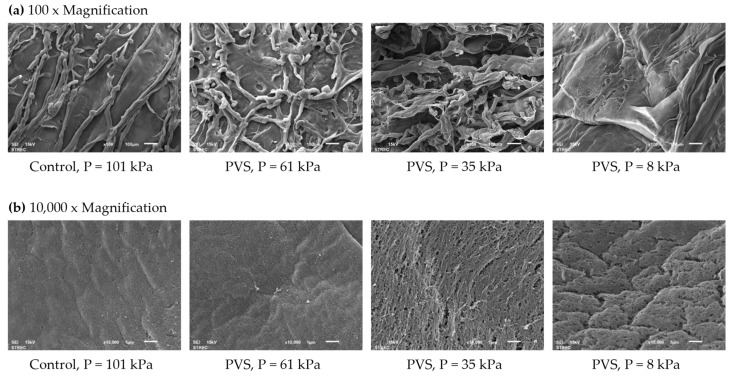
SEM images of the external structures of the calcium-alginate hydrogel balls as affected by different working pressures for 3 cycles at (**a**) 100 × Magnification and (**b**) 10,000 × Magnification (1 cycle of control = 60 min of soaking during the external gelation process, while 1 cycle of PVS = 10 min of depressurization and 10 min of vacuum liberation for 3 repetitions (60 min)).

**Figure 5 foods-10-01521-f005:**
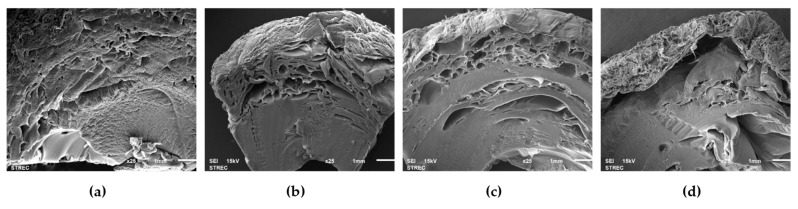
SEM images of the cross-sectional structures of the calcium-alginate hydrogel balls as affected by different working pressures for 3 cycles at 25 × magnification (1 cycle of control = 60 min of soaking during the external gelation process, while 1 cycle of PVS = 10 min of depressurization and 10 min of vacuum liberation for 3 repetitions (60 min)): (**a**) control, P = 101 kPa; (**b**) PVS, P = 61 kPa; (**c**) PVS, P = 35 kPa; and (**d**) PVS, P = 8 kPa.

**Table 1 foods-10-01521-t001:** Effect of working pressures and time on the diffusion coefficients (D) of calcium ion, rate of weight reduction (WR), and rate of shrinkage (S_v_) of calcium-alginate hydrogel balls.

Treatment	Pressure (kPa)	Time (Cycle ^1^)	D (×10^−4^ cm^2^/s)	*R* ^*2* 2^	Rate of WR (mg/s)	Rate of S_v_ (mm^3^/s)
Control	101	1	3.47 ± 0.38 ^cA^	0.993	0.43 ± 0.09 ^bAB^	0.37 ± 0.08 ^bA^
		2	0.51 ± 0.39 ^bA^	0.977	0.36 ± 0.06 ^bB^	0.32 ± 0.03 ^bB^
		3	0.26 ± 0.44 ^aB^	0.970	0.16 ± 0.07 ^aA^	0.12 ± 0.06 ^aB^
PVS	61	1	9.14 ± 1.73 ^cC^	0.980	0.48 ± 0.06 ^bAB^	0.38 ± 0.03 ^bA^
		2	0.83 ± 0.50 ^bB^	0.990	0.20 ± 0.07 ^aA^	0.31 ± 0.01 ^bB^
		3	0.49 ± 0.82 ^aC^	0.957	0.07 ± 0.12 ^aA^	0.00 ± 0.06 ^aA^
	35	1	5.17 ± 1.15 ^cB^	0.991	0.38 ± 0.05 ^bA^	0.31 ± 0.03 ^bA^
		2	1.86 ± 0.87 ^bC^	0.984	0.29 ± 0.04 ^bAB^	0.30 ± 0.04 ^bB^
		3	0.03 ± 0.03 ^aA^	0.975	0.12 ± 0.08 ^aA^	0.08 ± 0.04 ^aB^
	8	1	5.01 ± 0.95 ^cB^	0.964	0.54 ± 0.10 ^bB^	0.49 ± 0.05 ^cB^
		2	2.75 ± 1.47 ^bD^	0.952	0.18 ± 0.07 ^aA^	0.22 ± 0.04 ^aA^
		3	0.00 ± 0.00 ^aA^	0.976	0.39 ± 0.11 ^bB^	0.37 ± 0.05 ^bC^

D, diffusion coefficients. WR, weight reduction. Sv, volume shrinkage. ^1^ 1 cycle of control = 60 min of soaking during external gelation process, while 1 cycle of PVS = 10 min of depressurization and 10 min of vacuum liberation for 3 repetitions (60 min). ^2^
*R^2^* of the diffuion coefficient of calcium ion. Value is the mean ± standard deviation. Values within a column followed by different lowercase letters (time) and different uppercase letters (pressure within the time) differ by least significant difference (LSD) test (*p* < 0.05).

**Table 2 foods-10-01521-t002:** Linear regression ^1^ results showing the relation of weight reduction (WR) and volume shrinkage (S_v_) during the external gelation process of calcium-alginate hydrogel balls with and without pulsed-vacuum application.

Treatment	Pressure (kPa)	a	b	*R^2^*
Control	101	1.158 *	−3.192	0.988
PVS	61	0.861 *	3.651	0.967
	35	0.813 *	5.993	0.914
	8	0.851 *	5.024	0.975

^1^ S_v_ = a⋅WR + b. Significance level * indicates *p*-value of < 0.05.

**Table 3 foods-10-01521-t003:** Effect of working pressures on the textural characteristics of calcium-alginate hydrogel balls obtained at the pressurization time of 3 cycles ^1^.

Treatment	Pressure (kPa)	Hardness (N)	Breaking Deformation (%)	Young’s Modulus (kPa)	Rupture Strength (N)
Control	101	36.1 ± 2.3 ^a^	58.1 ± 0.3 ^a^	66.2 ± 2.7 ^a^	7.4 ± 0.6 ^ab^
PVS	61	51.0 ± 1.7 ^c^	63.0 ± 0.9 ^c^	95.0 ± 2.3 ^c^	7.0 ± 0.4 ^a^
	35	42.2 ± 2.1 ^b^	60.3 ± 1.1 ^b^	69.6 ± 6.4 ^b^	7.1 ± 0.5 ^a^
	8	64.5 ± 5.5 ^d^	66.9 ± 0.6 ^d^	142.7 ± 7.9 ^d^	8.2 ± 0.2 ^b^

^1^ 1 cycle of control = 60 min of soaking during the external gelation process, while 1 cycle of PVS = 10 min of depressurization and 10 min of vacuum liberation for 3 repetitions (60 min). Value is the mean ± standard deviation. Values within a column with different letters are significantly different (*p* < 0.05).

## Data Availability

This manuscript has no associated data.

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
