# Peer review of "The Feasibility of Using Pulsed-Vacuum in Stimulating Calcium-Alginate Hydrogel Balls"

_foods, 2021, doi:10.3390/foods10071521_

Round 1

Reviewer 1 Report

The manuscript is interesting and well prepared, however I recommend the minor revisions, as follows:

The practical aspects of analyzed materials should be provided

I recommend to add pictures of the analyzed materials

Line 40 please avoid the abbreviations

Statistical data should be revised. Usually the lowest values have „a” homogeneous group

Author Response

Response to Reviewer 1 Comments

Manuscript: foods-1254570

Title: The Feasibility of Using Pulsed Vacuum in Stimulating Calcium-Alginate Hydrogel Balls

We would like to thank the editor and reviewers for their helpful, constructive, and pertinent suggestions and comments, which we believe have helped to improve quality and scientific substance of the manuscript. The responses have been shown along the track change of the manuscript.

Point 1:  The practical aspects of analyzed materials should be provided.

Response 1: The practical aspect has been added in the sections of introduction and conclusion.

Point 2:  I recommend to add pictures of the analyzed materials.

Response 2: The images of analyzed materials have been provided in Figure 1.

Point 3:   Line 40 please avoid the abbreviations.

Response 3: The abbreviations have been removed.

Point 4:  Statistical data should be revised. Usually the lowest values have “a” homogeneous group.

Response 4: The statistical data has been revised in all tables.

Reviewer 2 Report

The manuscript is interesting scientific contributions to study on the feasibility of using pulsed vacuum in stimulating calcium-alginate hydrogel balls. In this regard, the present study was conducted with the objective to investigate the feasibility of using pulsed vacuum stimulation at different depressurization levels on the diffusion coefficients of calcium ion, textural and microstructural characteristics of Ca-Alg hydrogel balls.The paper has high scientific level, the experiment is well designed, the discussion is consistent and the final conclusions are interesting.

Suggestions for edition as well as some comments are the following:

Please change these keywords “pulsed vacuum; calcium-alginate” because they are presented in the title.

How many samples were elaborated?, please include this information in the manuscript

How many samples of each treatment were analysed?, please include this information in the manuscript

Why author selected these depressurization levels (8, 35, and 61 kPa)?

Please update or remove references before 2010.

Author Response

Response to Reviewer 2 Comments

Manuscript: foods-1254570

Title: The Feasibility of Using Pulsed Vacuum in Stimulating Calcium-Alginate Hydrogel Balls

We would like to thank the editor and reviewers for their helpful, constructive, and pertinent suggestions and comments, which we believe have helped to improve quality and scientific substance of the manuscript. The responses have been shown along the track change of the manuscript.

Point 1: Please change these keywords “pulsed vacuum; calcium-alginate” because they are presented in the title.

Response 1: The keywords have been changed as suggested.

Point 2: How many samples were elaborated?, please include this information in the manuscript.

Response 2: The number of samples per treatment and number of replications have been added
 throughout the manuscript.

Point 3: How many samples of each treatment were analysed?, please include this information in the manuscript.

Response 3: Similar to Response 2

Point 4: Why author selected these depressurization levels (8, 35, and 61 kPa)?

Response 4: Regarding to our preliminary test, the result showed that when the depressurization level higher than 61 kPa was applied, the quality attributes of Ca-Alg hydrogel ball was not difference from that hydrogel of atmospheric pressure. In addition, we cannot do the study at the pressurization level lower than 8 kPa due to the limitation of our machine.

Point 5:

Please update or remove references before 2010.

Response 5: We agree with the reviewer that the references before 2010 should be updated, however, some references are very relevant for explaining our studies. Therefore, kindly allow us to keep citing them.

Reviewer 3 Report

Dear Authors,

The manuscript is well-written and provides interesting results. Below, I indicate some suggestions and some questions.

Does ball volume have a significant role in the properties of hydrogel balls? Balls with 3 cm in diameter are suitable carrying systems for food applications?

Line 55: in vivo should be italicized

Line 367: Please remove “were”

Lines 363-364: Does 61 kPa PVS treatment could be recommended for further experiments? If so, please include a recommendation at the end of Conclusion section.

Line 371: “demanded less time”. The effect of time was not considered during alginate balls preparation. Please revise this statement.

Round 2

Reviewer 3 Report

Dear Authors,

The manuscript was properly revised.

Good job.